# Capturing Structural Locality in Non-parametric Language Models

**Frank F. Xu, Junxian He, Graham Neubig, Vincent J. Hellendoorn**
School of Computer Science
Carnegie Mellon University
{fangzhex,junxianh,gneubig}@cs.cmu.edu, vhellendoorn@cmu.edu

## Abstract

Structural locality is a ubiquitous feature of real-world datasets, wherein data points are organized into local hierarchies. Some examples include topical clusters in text or project hierarchies in source code repositories. In this paper, we explore utilizing this structural locality within *non-parametric language models*, which generate sequences that reference retrieved examples from an external source. We propose a simple yet effective approach for adding locality information into such models by adding learned parameters that improve the likelihood of retrieving examples from local neighborhoods. Experiments on two different domains, Java source code and Wikipedia text, demonstrate that locality features improve model efficacy over models without access to these features, with interesting differences. We also perform an analysis of how and where locality features contribute to improved performance and why the traditionally used contextual similarity metrics alone are not enough to grasp the locality structure.

## 1 Introduction

Language models (LMs) predict a probability distribution over sequences, and are most widely studied to model and generate natural languages (Bengio et al., 2003; Merity et al., 2018; Baevski & Auli, 2018; Brown et al., 2020). Advances in LMs benefit many natural language processing downstream tasks, such as machine translation (Bahdanau et al., 2015), dialog systems (Sordoni et al., 2015), question answering (Yang et al., 2019; Raffel et al., 2019), and general representation learning for natural language (Devlin et al., 2018; Liu et al., 2019). Recently, LMs have also been adopted to model sequences other than text, such as source code written in programming language (Hindle et al., 2016; Hellendoorn & Devanbu, 2017; Alon et al., 2020; Karampatsis et al., 2020), which can enable useful downstream tasks like code completion (Raychev et al., 2014).

Most current neural LMs are based on *parametric* neural networks, using RNN (Mikolov et al., 2010) or Transformer (Vaswani et al., 2017) architectures. These models make predictions solely using a fixed set of neural network parameters. Recently, more and more neural LMs also incorporate *non-parametric* components (Grave et al., 2017; Guu et al., 2018; He et al., 2020; Khandelwal et al., 2020), which usually first select examples from an external source and then reference them during the prediction. For example, Khandelwal et al. (2020) model the token-level probability by interpolating the parametric LM probability with a probability obtained from the nearest context-token pairs in an external datastore. Using such non-parametric components in LMs is beneficial because the model no longer needs to memorize everything about the language in its parameters.

For such non-parametric LMs, one important concept is a *distance* metric between the current context and other contexts in the datastore. One example of such metric is the $\ell^2$ distance between context vectors calculated by the parametric model (Khandelwal et al., 2020). This distance can be used in both retrieval and probability calculation; items in the datastore that are less distant from the current context are more likely to be retrieved and have a higher influence on the final probability. However, given that non-parametric datastores are typically very large, containing a myriad of contexts from disparate sources, calculating a metric that accurately reflects semantic similarities is non-trivial; as we demonstrate in experiments, there is much room for improvement in current practice.

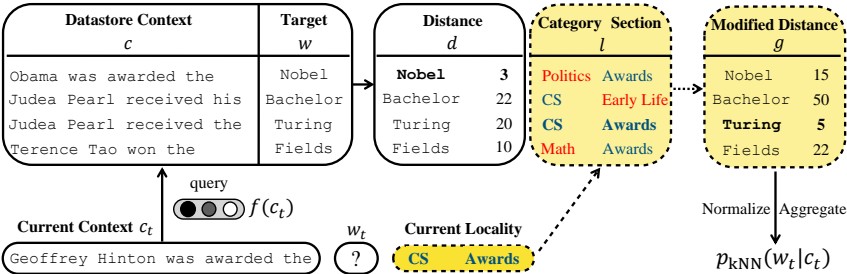

Figure 1: An example of incorporating structural locality in the computation flow of $p_{\text{kNN}}(w_t|c_t)$. The current context $c_t$ is used to calculate distance $d$ to contexts in the datastore $(c, w)$. Dashed boxes and lines represent components proposed in our work, which leverage structural information $l$ (non-local, local) to allow for more accurate modified distances $g$ (lower is more similar).

In this paper, we argue that the relevance of contexts may be correlated with not only contextual distance, but also structural characteristics of the underlying data. Specifically, we take advantage of a property we dub *structural locality*, the propensity of text to be divided into local groups sharing common hierarchical attributes. This property is ubiquitous across many kinds of texts and can provide additional information on how closely related two different examples are to each other. Throughout this paper, we will provide two case-studies of this phenomenon. First, in the domain of programs written in source code, if two source files originate from the same project, they are more likely to be related than files from other projects, and even more so if they are from the exact same package (Hellendoorn & Devanbu, 2017). Second, in natural language, two sections of Wikipedia text may be more related if they fall within the same topical domain, are from similarly titled sections, or even are from the same article (as in Figure 1). Notably this locality often manifests itself at different levels, such as the levels of "project", "subdirectory", and "file" cited above for source code.

In this paper, we hypothesize that by using multiple levels of structural locality, we can better calibrate the distance metrics used to retrieve examples from non-parametric datastores, thereby improving LM performance. Specifically, we propose a simple-yet-effective approach that can easily be applied to non-parametric LMs: we use different levels of structural locality to define functions that modify the contextual distance metrics used by the non-parametric module.

We evaluate our method on two drastically different domains: Java programming language source code, and natural language Wikipedia articles, achieving noticeable LM performance gains in both by adding just 5 & 7 parameters respectively. Moreover, we perform an in-depth analysis showing how the traditionally used contextual similarity metrics alone are not enough to grasp the locality structure, providing evidence for why adding the locality features is indeed useful. We also compare programming languages and natural languages to highlight several interesting differences in terms of how, and how much, the locality helps improve LM performance.

## 2 NON-PARAMETRIC LANGUAGE MODELS

Given a linguistic context consisting of a sequence of tokens $c_t = (w_1, ...w_{t-1})$, autoregressive parametric LMs estimate $p(w_t|c_t; \theta)$, the probability distribution over the next token $w_t$. Such parametric LMs store information regarding the language being modeled in the parameters $\theta$. The size of $\theta$ is fixed in advance based on the hyperparameters of the model architecture, in recent years typically a neural network (Grave et al., 2016; Baevski & Auli, 2018; Dai et al., 2019; Brown et al., 2020). In contrast, a non-parametric LM's number of parameters is not determined by just the model architecture, but also by the underlying data used to train the model. While non-parametric LMs using Bayesian statistics have existed for some time (Wood et al., 2011; Shareghi et al., 2017; He et al., 2020), they have recently seen increased prevalence through the introduction of neural LMs that retrieve relevant examples from an external datastore (Hashimoto et al., 2018; Guu et al., 2018). In particular, we focus on kNN-LMs (Khandelwal et al., 2020), a variety of such models that uses a nearest neighbor retrieval mechanism to augment a pre-trained parametric LM, achieving impressive results without any additional training.

Neural network-based LMs usually map the context $c$ to a fixed-length vector representation, with a trained function $f(c)$. In kNN-LMs, the non-parametric component consists of a collection ($\mathcal{D}$) of contexts for the kNN to retrieve from. Denoting these contexts and their corresponding next token as

Table 1: Locality features designed for each data type according to domain knowledge.

| Locality | Wikipedia text | Java projects |
|---|---|---|
| $l_0$ | different article category, different section title | different project |
| $l_1$ | same article category, different section title | same project, different subdirectory |
| $l_2$ | same section title, different article category | same subdirectory |
| $l_3$ | same section title, same article category | – |

$(c_i, w_i) \in \mathcal{D}$, we create a datastore $(\mathcal{K}, \mathcal{V}) = \{(k_i, v_i)\}$, which contains key-value pairs:

$$(\mathcal{K}, \mathcal{V}) = \{(f(c_i), w_i) \mid (c_i, w_i) \in \mathcal{D}\} \tag{1}$$

During inference, the parametric component of the LM generates the output distribution over next tokens $p_{LM}(w_t|c_t; \theta)$ and the corresponding context representation $f(c_t)$, given the test input context $c_t$. Then the non-parametric component of the LM queries the datastore with $f(c_t)$ representation to retrieve its $k$-nearest neighbors $\mathcal{N}$ according to a distance function $d(\cdot, \cdot)$. We can then compute a probability distribution over these neighbors using the softmax of their negative distances. The model aggregates the probability mass for each vocabulary item across all its occurrences in the retrieved targets. This distribution is then interpolated with the parametric LM distribution $p_{\text{LM}}$ to produce the final kNN-LM distribution:

$$p_{\text{kNN}}(w_t|c_t) \propto \sum_{(k_i, v_i) \in \mathcal{N}} \mathbf{1}_{w_t = v_i} \exp(-d(k_i, f(c_t))) \tag{2}$$

$$p(w_t|c_t; \theta) = \lambda p_{\text{kNN}}(w_t|c_t) + (1 - \lambda) p_{\text{LM}}(w_t|c_t; \theta) \tag{3}$$

In our experiments, we follow Khandelwal et al. (2020) in setting the interpolation factor $\lambda$ to 0.25.

## 3 DEFINING STRUCTURAL LOCALITY

We define structural locality as a categorical feature calculated between a pair of contexts $(c_i, c_j)$ in a collection of data, that describes whether the pair share some common, potentially hierarchical, attributes (e.g., the section title of a Wikipedia article section, or the directory path of a source code file). For each domain, a set of hierarchical attributes $\{l_0, l_1, ..., l_n\}$ can be defined based on prior knowledge of the domain. We denote $l_k(c_i, c_j) \in \{0, 1\}$ as the boolean *locality feature* value for the context pair, representing whether $c_i$ and $c_j$ share the same hierarchical attributes $l_k$. Here, $l_0$ is reserved for "no locality", in case the pair shares none of the attributes. Without loss of generality, we set a constraint that $\sum_k l_k(c_i, c_j) = 1$, as new features can be introduced by conjunction and negation of the attributes if needed.

**Specific Instantiations.** We instantiate these features on our two case studies of Wikipedia text and Java source code, as summarized in Table 1.

In Wikipedia, for every context $c_i$, we define four mutually exclusive hierarchical attributes, $l_0 - l_3$. We calculate these features based on the Wikipedia article and section titles, using simple pattern matching. We then link each article to a set of categories (one article may belong to multiple categories) using the knowledge graph WikiData,[1] by aggregating all the category entities involving two properties: P31 (instance of) and P279 (subclass of). The criterion for "same section title" is exact string match (Hayashi et al., 2020). If there is at least one common category between the sets of categories for two articles, the pair is assigned the "same article category".

For Java source code, we define 3 mutually exclusive attributes, $l_0 - l_2$ based on the location of the code. For each source file, we use the full file path to obtain the two attributes: project name and sub-directory path.[2] The criterion for both "same project" and "same subdirectory" is exact string match. Note that these features are strictly hierarchical, hence only two features are used to capture specific locality here.

**An Aside: Connections to Domain Adaptation.** Domain adaptation typically refers to reusing existing information about a given problem (e.g., data or model) to solve a task in a new domain.

---

[1] https://www.wikidata.org/

[2] For example, full path `.../Journal.IO/src/main/java/journal/io/api/DataFile.java` has project `Journal.IO` and sub-directory `src/main/java/journal/io/api/` for package `journal.io.api`.

Domain adaptation for neural models generally focuses on fine-tuning models on in-domain data (Sennrich et al., 2016; Chu et al., 2017) or making direct modifications to the model to consider domain information (Britz et al., 2017) or latent topic features (Khudanpur & Wu, 2000; Mikolov & Zweig, 2012; Wang & Cho, 2016). Most of these methods do not natively support new test-time contexts that were not seen at training time. In comparison, one immediate advantage of non-parametric LMs is the ability to adapt to different domains at test time without re-training (Merity et al., 2016; Grave et al., 2016; 2017; Khandelwal et al., 2020). For example, some adaptive LMs (Grave et al., 2016; 2017) make use of the previous hidden states of test documents dynamically during inference. Similarly, our proposed locality features do not require re-training on the training set.

Note that within the scope of this paper, although connected, the proposed *structural locality* is a different concept from *domain*. We consider domains as higher-level classifications describing the text where one example belongs to one domain label; e.g., a section about Kim Kardashian's early life belongs to a category of texts describing celebrities. One the other hand, with the structural locality, a user could define multiple levels of locality: to that same section, we can assign not only the domain label, but also, the section title "Early Life". The lightweight nature of our model combined with non-parametric LMs also makes adding more levels of features straightforward, as the features only need to be calculated for the top nearest neighbors, and the number parameters that need tuning in our proposed method (Section 5) is only about twice the number of locality features.

## 4 STRUCTURAL LOCALITY AND NEAREST NEIGHBORS

In this section, we examine the relationship between distances derived from neural LM features $d(f(c_i), f(c_t))$, structural locality features $l(c_i, c_t)$, and the accuracy of the next-word prediction $w_i$. Specifically, the underlying assumption of the kNN-LM is that less distant contexts will be more likely to accurately predict the next word $w_t$. We would like to test whether this correlation between distance $d(\cdot)$ holds uniformly across different locality levels $l(\cdot)$, or if locality provides additional information indicative of whether a particular context is useful for predicting $w_i$ beyond just that provided by the neural representations.

**Data.** We use two different corpora from different domains to examine this question.

WIKITEXT-103[3] is a standard language modeling benchmark (Merity et al., 2016) consisting of natural language text from English Wikipedia. It contains a 250K token, word-level vocabulary, with 103M tokens in the training set and 250K tokens in both the validation and test sets.

JAVA GITHUB[4] is a programming language corpus containing Java source code from Github (Allamanis & Sutton, 2013) that is widely used in source code modeling (Hellendoorn & Devanbu, 2017; Karampatsis et al., 2020). It contains 1.44B tokens from 13,362 projects in the training split, 3.83M tokens from 36 projects in the validation split and 5.33M tokens from 38 projects in the test split. The splits are separated by whole projects. The dataset is tokenized with byte-pair encoding (Sennrich et al., 2015) using the vocabulary from Karampatsis et al. (2020) with 2,000 subtokens.

**Base Model.** As the neural model used to calculate context features, we follow Khandelwal et al. (2020),[5] train an LM with the exact architecture and optimization described by Baevski & Auli (2018): a decoder-only Transformer (Vaswani et al., 2017), with 1024 dimensional hidden states for the WIKITEXT-103 dataset and 512 for JAVA GITHUB. We set the number of retrieved nearest neighbors to be analyzed to 1024, and the distance metric to $\ell^2$ following the default.

**Datastore.** To capture the effect of our proposed locality features, the datastore should ideally be both closely related to the test examples, sufficiently large to ensure precise kNN retrieval performance for a wide range of contexts, and not too sparse in terms of the prevalence of locality features.

For WIKITEXT-103, we include the training set, as well as the validation/test set (excluding the text currently being evaluated) in the datastore. For the JAVA GITHUB, due to the relatively large size of the validation/test set, and the unwieldy size of the training set, we include only the validation/test set (also excluding the current file).

---

[3] https://blog.einstein.ai/the-wikitext-long-term-dependency-language-modeling-dataset/.

[4] https://zenodo.org/record/3628665.

[5] https://github.com/urvashik/knnlm

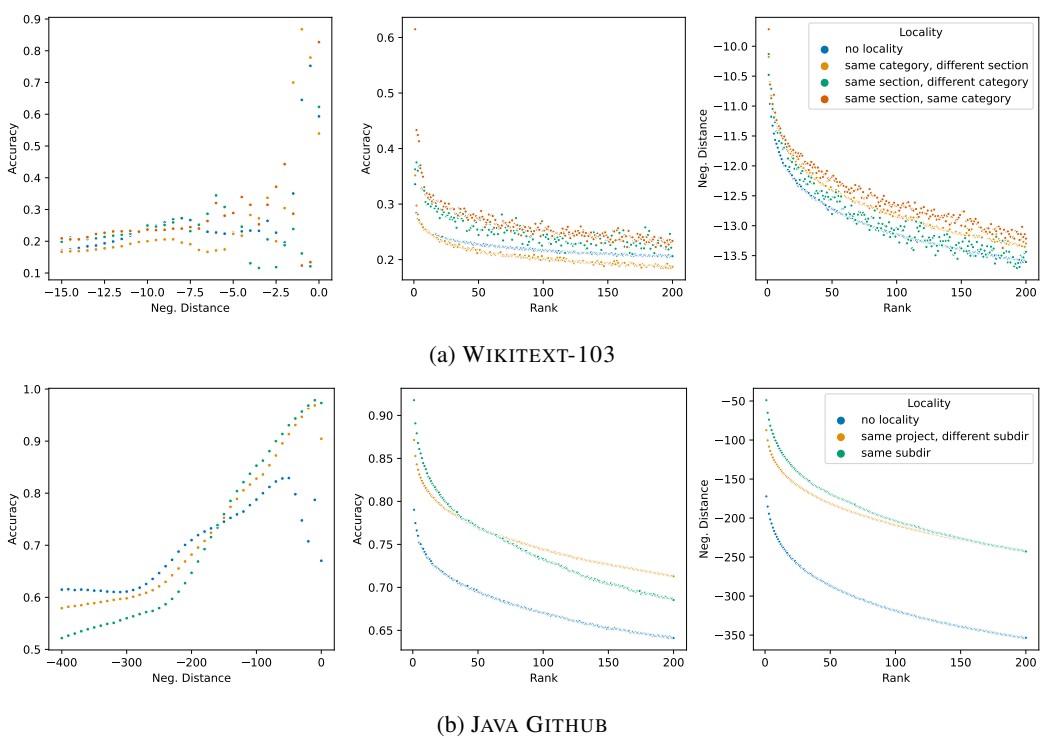

(a) WIKITEXT-103

(b) JAVA GITHUB

Figure 2: The relationship between nearest neighbor rank, negative distance, and the retrieval accuracy, grouped by different locality levels. Only top 200 nearest neighbors are shown for clarity. Negative distance on x-axis represents the upper bound of the bin.

**Analysis.** Consider $k$ nearest neighbor contexts $\mathcal{N}_t = \{c_r | r = 1...k\}$ retrieved for any test context $c_t$ in the test set $\mathcal{C}$, ordered by the ascending distance: $\forall r : d(c_r, c_t) < d(c_{r+1}, c_t)$. We define $r \in [1, k]$ as the "rank" for the retrieved context $c_r$. To study the quality of the retrieved contexts, we calculate the number of correctly retrieved tokens, defined as $\#\{w_r = w_{t_{\text{gold}}}\}$ across $\mathcal{C}$.

We plot in Figure 2, from left to right: (1) Negative distances $\{-d(c_r, c_t) | c_r \in \mathcal{N}_t, c_t \in \mathcal{C}\}$ grouped into bins, vs. the retrieval accuracy of the bin $avg(\#\{w_r = w_{t_{\text{gold}}}\})$. (2) Rank $r$ vs. the retrieval accuracy at rank $r$, $avg(\#\{w_r = w_{t_{\text{gold}}}\})$. (3) Rank $r$ vs. the average negative distance $avg(-d(c_r, c_t))$ at rank $r$. All of the plots are grouped by different locality levels $l_0$ to $l_n$.

Naturally, the left-most sub-figures reflect that the lower the (negative) distance, the lower the accuracy on both datasets. Yet, interesting, on the Wikipedia domain (Figure 2a), as the negative distance gets close to 0 (perfect match), the retrieval accuracy for the next word does not always increase; the accuracy values in this range have very high variance and all 4 levels of locality show no clear trend. This partly indicates that context-based distance is imperfect, regardless of locality. Even so, at slightly lower distances, the trends stabilize and largely show a consistent picture: more specific locality features, especially those involving the same category ($l_1 \& l_3$) yield better predictions than the locality-insensitive component for identical distances. This is especially significant at higher ranked retrievals (middle sub-figure), where contexts that share the same section title and the same article category are substantially more likely to share the same completion. This suggests that the proposed locality features are not fully represented by, or correlated with the original distance metric, and thus implies that there is room for improvement by incorporating these features.

In the Java source code domain (Figure 2b), we generally observe that the retrieval accuracy is much higher than that in the Wikipedia domain, suggesting that the kNN is doing a better job retrieving relevant contexts. This is largely due to higher repetitiveness of source code (Hindle et al., 2016); as we show later, the base Transformer model also performs much better here than on natural language text (Section 6.2). We also observe a more pronounced locality effect here: at the same distances close to 0 and at the same rank, neighbors that are local to the current context have far higher accuracy, indicating usefulness of locality features in the source code domain as well. However, as we can see

from the (right-most) plot of rank versus the negative distance, the average distances of the neighbors with higher locality levels are also significantly smaller than the distance of those without locality. This suggests that the distance in the Java source code domain already correlates well with the level of locality, which may render incorporating locality features less beneficial. We study the precise benefit under one instantiation of this model next.

## 5   INCORPORATING STRUCTURAL LOCALITY IN NON-PARAMETRIC LMS

Now that we have demonstrated that locality is additionally indicative of next-word prediction accuracy beyond context distance, we propose a method to incorporate this information into the non-parametric retrieval module. In the case of kNN-LMs (Section 2), recall that $p_{\text{kNN}}$ is calculated based on the softmax of the negative distance $-d(f(c_i), f(c_t))$. Assuming locality features $\{l_0, l_1, ..., l_n\}$ for each pair $(c_i, c_t)$ consisting the retrieved nearest neighbor and the current inference context $c_t$, we modify the formulation of $p_{\text{kNN}}$ (Equation 3) to consider these features as below:

$$p_{\text{kNN}}(w_t|c_t; \{\theta_n\}) \propto \sum_{(k_i, v_i) \in \mathcal{N}} \mathbf{1}_{w_t = v_i} \exp(-g(k_i, c_t; \{\theta_n\})) \tag{4}$$

$$g(k_i, c_t; \{\theta_n\}) = g_n(d(k_i, f(c_t)); \theta_n) \text{ if } l_n(c_i, c_t) = 1. \tag{5}$$

where $g_n(d(\cdot, \cdot); \theta_n)$ is a learnable function of the distance of the nearest neighbors, with parameter $\theta_n$ for each type of locality feature $l_n$. One can view function $g(\cdot)$ as a "modified" distance for nearest neighbors after taking locality information into consideration. In our experiments, we adopt a linear form of $g(\cdot)$:

$$g_n(d(\cdot, \cdot); w_n, b_n) = w_n d(\cdot, \cdot) + b_n \tag{6}$$

We omit the bias for $g_0(\cdot)$ by setting $b_0 = 0$ to remove one free parameter from the model and potentially make optimization easier.[6] To learn these functions, a user needs to provide just a small sample of annotated data in the same domain, as there are only $2n + 1$ parameters to optimize. In our experiments, we use the validation split for optimization. The parameters are trained to minimize the negative log-likelihood of the kNN prediction of the gold token:

$$\underset{\{\theta_n\}}{\arg\min} -\log p_{\text{kNN}}(w_t = w_{t_{\text{gold}}}|c_t; \{\theta_n\}) \tag{7}$$

To optimize the parameters, we use the Adam (Kingma & Ba, 2014) optimizer with a learning rate of 0.0001 on the validation set for 200 epochs. It converges within 20 minutes for both datasets.

## 6   HOW DOES STRUCTURAL LOCALITY IMPROVE LANGUAGE MODELING?

### 6.1   EXPERIMENTAL SETUP

**Baselines.** Since we base our model on kNN-LMs, this model will be our most directly comparable baseline. We also compare our model to the underlying parametric LM (Baevski & Auli, 2018), without the kNN module. For the JAVA GITHUB dataset, we additionally compare to the recent state-of-the-art model from Karampatsis et al. (2020) on code language modeling, which uses BPE and LSTMs. In all experiments, the maximum number of tokens per input sample is 3,072.

**Evaluation.** We evaluate the performance of the LM with the standard perplexity metric and token prediction top-$k$ accuracy on the held-out data.[7] The top-$k$ accuracy is calculated by checking if the ground truth token is among the predicted top-$k$ list. This metric, primarily for $k \in \{1, 5\}$ (with more $k$ values in Appendix A.2), is commonly used to evaluate predictive models of source code (Hindle et al., 2016). In order to more easily incorporate and analyze the locality features, and also following Karampatsis et al. (2020), we split the evaluation dataset into independent test examples to evaluate, where each of the example is an atomic unit in the locality hierarchy. For JAVA GITHUB, each test example is a source code file, and for WIKITEXT-103, each example is an article section.[8]

---

[6]We also experimented with an adaptive variant that conditioned the weights and biases ($\{w_n\}, \{b_n\}$) on the current context representation $f(c_t)$ parameterized by a MLP. However, this did not result in significant improvement over directly optimizing $w$ and $b$ (Appendix A.3).

[7]For JAVA GITHUB, the perplexity is calculated on full tokens by aggregating the likelihood of subtokens. The accuracy is calculated that *all* subtokens in a full token have to be predicted correctly.

[8]Note that because we predict WIKITEXT-103 section-by-section instead of article-by-article the perplexity numbers reported here are somewhat worse than other works. Article-by-article calculation is not inherently

Table 2: Perplexity and top-$k$ token prediction accuracy results on two datasets. *Uses released pre-trained model, †no stochastic training, for all others stddev $< 0.01$ for 5 runs.

| Dataset | Model | Dev PPL | Test PPL | Rel. Gain | Top-1 Acc. (Rel. Err. Red.) | | Top-5 Acc. (Rel. Err. Red.) | |
|---|---|---|---|---|---|---|---|---|
| WIKITEXT -103 | Transformer[1] | *23.31 | *23.73 | – | 39.0% | (–) | 64.0% | (–) |
| | +kNN[2] | †20.21 | †19.94 | 16.0% | 41.3% | (3.79%) | 66.8% | (7.91%) |
| | +kNN + locality | **19.51** | **19.16** | 3.9% | **43.2%** | (3.29%) | **68.0%** | (3.56%) |
| JAVA GITHUB | BPE LSTM[3] | – | *5.27 | – | – | | – | |
| | Transformer | 3.29 | 3.07 | 41.7% | 75.6% | (–) | 87.6% | (–) |
| | +kNN | †2.43 | †2.18 | 29.0% | 83.9% | (34.0%) | 96.0% | (67.9%) |
| | +kNN + locality | **2.37** | **2.13** | 2.3% | **84.7%** | (4.91%) | **96.6%** | (15.0%) |

[1]Baevski & Auli (2018), [2]Khandelwal et al. (2020), [3]Karampatsis et al. (2020)

## 6.2 RESULTS

The high-level results are shown in Table 2. At first glance, we can already see that the two datasets vary greatly in predictability. With a similar Transformer architecture, performance on JAVA GITHUB is much better than on the WIKITEXT-103 across the board, partly due to the rigid nature of programming language syntax. With a Transformer model, we achieved a strong state-of-the-art language model on Java code, with low perplexity and very high prediction accuracy (~75%).

By adding kNN module onto the Transformer-based LMs, perplexity and accuracy in both domains improves by a large margin. This is expected and in line with previous experiments on kNN-LMs (Khandelwal et al., 2020). The Wikipedia domain enjoys less relative improvement in perplexity (16%) than the Java source code domain (29%). This is particularly interesting, considering that the datastore used for WIKITEXT-103 contains both the current held-out split and the training data (~100M contexts), compared to that for JAVA GITHUB with only the current held-out split (~5M contexts). This reflects the fact that source code is known to benefit strongly from project- and package-specific locality (Tu et al., 2014; Hellendoorn & Devanbu, 2017).

Adding proposed locality features and finetuning the parameters on the validation set improves the performance further on both datasets, albeit with a smaller relative gain. This confirms our hypothesis that incorporating locality into the non-parametric retrieval-based LMs is beneficial. We also see that locality features in the Wikipedia domain result in fairly consistent gains, while the Java source code domain sees especially strong accuracy improvements. This echoes our analysis of the source code corpus in Section 4, where we found that distance was generally

Table 3: Learned parameters $\theta_0, \{\theta_n\}$ for each locality level and a non-local level $g_0$.

| | WIKITEXT-103 | | JAVA GITHUB | |
|---|---|---|---|---|
| | $w$ | $b$ | $w$ | $b$ |
| $g_0$ | 1.233 | – | 0.022 | – |
| $g_1$ | 1.246 | -1.087 | 0.033 | -3.627 |
| $g_2$ | 1.288 | -1.250 | 0.041 | -5.920 |
| $g_3$ | 1.285 | -1.464 | – | – |

strongly correlated with accuracy, but that locality was particularly informative at low distances. There, it may help discriminate between top-ranked completion candidates (as also shown later in Tab. 4). It is notable that despite the fact that the perplexity and accuracy on JAVA GITHUB are already very strong with the vanilla Transformer, we still see a noticeable relative error reduction of 4.9% by adding locality levels information.

We next study how locality features guide towards a "better" distance distribution among nearest neighbors. We plot the relationship between the nearest neighbor ranks and "modified" distance $g(k_i, c_t)$ in Figure 3. Table 3 shows the specific learned parameters for each level of $g(\cdot, \cdot)$. Evidently, the biases and weights vary accordingly with locality levels, as the model tries to "correct" the original distance by emphasizing more local contexts. Compared with the original negative distance $-d(k_i, f(c_t))$ depicted in Figure 2, the negative modified distance is more separated between the different locality levels on either dataset, showing the relative importance of different locality more clearly. We analyze several alternative approaches to parameterizing locality in Appendix A.3.

For WIKITEXT-103, comparing Figure 3a with Figure 2a, we can see that with the original distance different localities cluster together, and the modified distance separates them much better. We can also

---

incompatible with our proposed method, but it would require additional implementation to use different locality features for different locations in the output. Hence, we used section-by-section calculation for expediency.

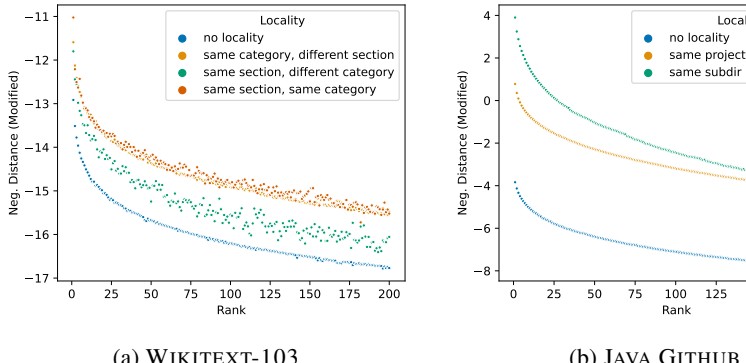

(a) WIKITEXT-103                    (b) JAVA GITHUB

Figure 3: The relationship between nearest neighbor rank and the "modified" negative distance $-g$ guided by our proposed locality features, grouped by different locality levels. Similarly to Figure 2, only top 200 nearest neighbors are shown for clarity.

see that if two contexts have the same article category and the same section title, then their distance on average is the closest, closely followed by those sharing article categories only. On the other hand, contexts that only share section titles are not as closely related. This is intuitively reasonable; the education section for a computer scientist and a musician can be very different in content, even if sharing some commonalities, like the phrases used to describe courses, grades, school locations, etc. This also underscores the usefulness of explicit locality features in providing interpretable insights into the relevance of domain knowledge.

For JAVA GITHUB, comparing Figure 3b with Figure 2b, we can see that the original distance is already more separated between different locality levels than that of WIKITEXT-103, again suggesting better learned representations (in terms of locality sensitivity) for the Java domain. However, the model still benefits somewhat from the contexts that are under the same subdirectory (more so than just the same project), especially for top nearest neighbors: the gap for ranks higher than 80 is more pronounced with the modified distance. This again verifies our hypothesis about the hierarchical nature of structural locality. It also indicates potential practical applications – if this model were deployed in a code editor, one could obtain representations of the files in the same sub-directory as the current file and use them, along with the proposed locality features, to bias auto-complete results.

Table 4 shows a randomly sampled test context from each domain where $p_{kNN}$ for the gold token increases after using locality features. We can see that the nearest neighbor search using context representations performs reasonably well at capturing patterns and themed phrases, especially closer to the last token, finding two very similarly rated candidates. However, in both examples, the second retrieved candidate has a wrong target token. Informed by locality features – in the WIKITEXT-103 example, a matching section *and* category for the first candidate – the more "local" context enjoys a large boost in probability, while the non-local one's decreases slightly. We present additional examples in Appendix A.1. The JAVA example demonstrates the same effect; the second retrieved example shows resemblances in variable name and template structures, but the fact that the project is focused on Google API rather than Twitter API makes the original retrieval undesirable.

## 7   CONCLUSION

In this paper, we propose a novel[9] method of incorporating structural locality into non-parametric LMs that reference retrieved examples from a datastore. We evaluate this approach in both a natural language and programming language domain, and empirically explore the similarities and differences of how structural locality affects LMs in these settings. The improvement in perplexity and prediction

---

[9]Previous work (Hellendoorn & Devanbu, 2017) made the observation that source code files from the same GitHub repository or sub-directory tend to be relatively similar, but did not include an empirical analysis of this effect. Rather, their observation was backed up by improved performance of their $n$-gram language model with multiple tiered caches. Our work gives more fine-grained insights into this phenomenon, expands the applicability to neural models and new domains, and proposes a more generalized formulation for encoding multiple localities across multiple domains. See Appendix A.4 for a detailed discussion of the connection to previous work and novelty.

Table 4: Examples from two domains where incorporating locality features (non-local, local) lead to a significant increase in the cumulative $p_{\text{kNN}}$ for the gold token, with corresponding change in probability (normalized negative distance) for two nearest neighbors.

| Test Context | Test Target | Initial $\log p_{\text{kNN}}$ | $\Delta$ $\log p_{\text{kNN}}$ |
|---|---|---|---|
| Section: Seasonal forecasts; Category: Pacific typhoon season *The forecast indicated the potential for 26.2 tropical storms, compared to the 10– and 30-year average of 27.8 and 26.3 storms, respectively. The following month, the group raised their ...* | forecast | -2.20 | +0.89 |

| Datastore Context | Datastore Target | Orig. Log-Prob. | $\Delta$Log-Prob. |
|---|---|---|---|
| Section: Seasonal forecasts; Category: Pacific typhoon season *Their main reasons behind this is due to weaker trade force winds occurring in many parts of the basin, and there would be an enhanced cyclonic vorticity over the northwestern part of the Pacific. On April 27, the GCACIC made their first ...* | forecast | -2.91 | +1.25 |
| Section: Earthquake; Category: earthquake *Nickson Sioni from Simbo came on the (HF) radio and announced the arrival of a huge wave that had washed away several houses and come inland about 200m. This information was passed on by telephone to the Hawaii-based Pacific Tsunami Warning Center who then upgraded their ...* | warning | -3.01 | -0.31 |

| Test Context | Test Target | Initial $\log p_{\text{kNN}}$ | $\Delta$ $\log p_{\text{kNN}}$ |
|---|---|---|---|
| Directory: .../android/twitter/AuthConstants.java; Project: twitterdroid `public static final String CONSUMER_SECRET = "YOUR_CONSUMER_SECRET"; public static final String REQUEST_URL = "http://www. ...` | twitter | -2.03 | +0.49 |

| Datastore Context | Datastore Target | Orig. Log-Prob. | $\Delta$Log-Prob. |
|---|---|---|---|
| Directory: .../jtwitter/TwitterConnection.java; Project: twitterdroid `public static final String FRIENDS_TIMELINE_URL = "http://api.twitter.com/1/statuses/friends_timeline.xml"; public static final String UPDATE_URL = "http://api. ...` | twitter | -1.99 | +0.17 |
| Directory: .../impl/ActivityTemplate.java; Project: spring-social-google `private static final String ACTIVITIES_PUBLIC = "/activities/public"; private static final String ACTIVITIES_URL = "https://www. ...` | googleapis | -1.87 | -0.09 |

accuracy across both domains show the effectiveness and ubiquity of such locality information. Besides language modeling, we also envision that the method can benefit other applications that could be enhanced using user-defined prior domain knowledge such as conditional generation or representation learning using retrieved information.

**Limitations.** Our method applies to settings where locality effects are present, there is sufficient data to build a reliable datastore for each locality level, and that locality is not already meaningfully captured by the model. While this may not apply to every domain, these features are common: besides source code & Wikipedia, domains including books (features: authorship & dewey decimal system information), research papers (venue, research area), product manuals (kind, sections), and online discussions (time, topic) are all plausible candidates. The features in our studied domains were selected based on availability and prior knowledge of the domain (e.g., for Java, Hellendoorn & Devanbu (2017)). While they did provide measurable improvements and were natural to interpret, these may not be the optimal choice, and other options are worth investigating. It is also possible that LM improvements will eventually lead to learned context representations that almost perfectly capture the relevant locality information. However, we believe this to be unlikely: in many practical settings, there is some inherent ambiguity in partial contexts that cannot be solved with the surface text only. For instance, in Java source code files, it is common to declare a package, which will obviously match perfectly based on the first few tokens (e.g., `package org.`) with many other contexts. Yet given the scoped nature of this declaration, locally retrieved continuations are inherently far more useful.

## ACKNOWLEDGEMENTS

We thank Uri Alon for the helpful discussions and thorough feedback. We also thank the reviewers for the helpful conversations during the revision period of the paper. This work was supported in part by the National Science Foundation under Grant #1815287, and a gift from Amazon AWS.

## ETHICS STATEMENT

There are several ethical considerations regarding our proposed method. First, while language models have a large variety of positive uses in generating natural language (Li et al., 2021) or source code (Allamanis et al., 2018), there is also a potential for dual use. For example, previous work has cited the potential to generate fake news (Zellers et al., 2019) or undesirable/defamatory content (Wallace et al., 2019). Our methodology improves the accuracy of language models, which has the potential to increase their applicability not only in positive use cases but also in ethically questionable scenarios. In order to mitigate these risks, methods to detect machine generated content may be employed, although these are not perfect remedies (Zellers et al., 2019).

Second, because our methodology explicitly references the training corpus in generation it may increase the likelihood of copying content more-or-less verbatim from the training text. This raises potential issues of copyright violation (Chen et al., 2021) or privacy violation (Carlini et al., 2021). However, at the same time, because non-parametric models increase traceability through direct references to the training corpus, it also provides a tool to identify the provenance of the original content, providing a tool to identify such verbatim copying compared to standard parametric models.

Finally, there has been much recent interest in the energy and environmental impact of large language models (Strubell et al., 2019). Due to the necessity to look up information in a datastore, non-parametric models have additional computational overhead compared to parametric models. However, at the same time, as noted by Khandelwal et al. (2020), non-parametric models also provide a tool for quickly adapting to new domains through the use of domain-specific datastores, obviating the necessity for domain-specific fine-tuning. Our work takes this a step further, allowing models to leverage locality of the datastores, potentially making this an even more attractive alternative for efficient adaptation.

## REPRODUCIBILITY STATEMENT

The source code package containing a README document on how to reproduce the results and analysis and experiment scripts is available in the paper's supplementary material. The details about the dataset used, model hyper-parameters, and analysis performed are described in Section 4 and Section 6.1. All experiments are conducted on a single machine with a 48 core CPU and 8 NVIDIA V100 32GB GPU. For WIKITEXT-103 we use the pretrained model provided by (Khandelwal et al., 2020) for fair comparison. For JAVA GITHUB the Transformer model is trained until it converges, requiring approximately 2 days. The datastore size is about 5GB for JAVA GITHUB and 351GB for WIKITEXT-103.

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

## A APPENDIX

### A.1 ADDITIONAL EXAMPLES

We provide additional examples on WIKITEXT-103 in Table 5.

| Test Context | Test Target | Initial $\log p_{\text{kNN}}$ | $\Delta \log p_{\text{kNN}}$ |
|---|---|---|---|
| Section: Design; Category: ship class *In an effort to outmatch the American New York class, planners called for a ship armed with twelve 14-inch (36 cm) guns and faster than the 21 knots (39 km/h; 24 mph) of their rivals. Vickers files show that the Japanese had access to the designs for double- and triple-gun turrets, yet opted for six double turrets over four triple turrets. The final design—designated A-64 by the IJN—called for a ...* | displacement | -2.54 | +1.09 |

| Datastore Context | Datastore Target | Orig. Log-Prob. | $\Delta$Log-Prob. |
|---|---|---|---|
| Section: Design; Category: ship class *Both ships were also given torpedo bulges to improve their underwater protection and to compensate for the weight of the additional armour. In addition, their sterns were lengthened by 7.62 metres (25 ft). These changes increased their overall length to 213.8 metres (701 ft), their beam to 31.75 metres (104 ft 2 in) and their draft to 9.45 metres (31 ft). Their ...* | displacement | -3.25 | +1.23 |
| Section: History; Category: gun mount *The British Admiralty ordered a prototype of Coles's patented design in 1859, which was installed in the ironclad floating battery, HMS Trusty, for trials in 1861, becoming the first warship to be fitted with a revolving gun turret. Coles's aim was to create a ...* | ship | -2.98 | -0.24 |

| Test Context | Test Target | Initial $\log p_{\text{kNN}}$ | $\Delta \log p_{\text{kNN}}$ |
|---|---|---|---|
| Section: La Venta; Category: colossal statue *When discovered it was half-buried; its massive size meant that the discoverers were unable to excavate it completely. Matthew Stirling fully excavated the monument in 1940, after clearing the thick vegetation that had covered it in the intervening years. Monument 1 has been ...* | moved | -2.97 | +1.22 |

| Datastore Context | Datastore Target | Orig. Log-Prob. | $\Delta$Log-Prob. |
|---|---|---|---|
| Section: San Lorenzo; Category: colossal statue *The sculpture suffered some mutilation in antiquity, with nine pits hollowed into the face and headdress. San Lorenzo Colossal Head 10 (also known as San Lorenzo Monument 89) has been ...* | moved | -4.18 | +1.36 |
| Section: San Lorenzo; Category: castle *The excavations investigated the north of the fortress, searching for an entrance postulated by architect Eugene Viollet-le-Duc, but no such entrance was found. However, the excavation did reveal was that there was an addition to the north of the castle to enable the use of guns. Typologically, the structure has been ...* | dated | -4.63 | -0.11 |

Table 5: Additional WIKITEXT-103 examples where incorporating locality features (non-local, local) lead to a significant increase in the cumulative $p_{\text{kNN}}$ for the gold token, with corresponding change in probability (normalized negative distance) for two nearest neighbors.

### A.2 ADDITIONAL RESULTS ON TOKEN PREDICTION ACCURACY

We show additional results on top-$k$ ($k = 10, 20$) accuracy and relative error reduction (RER) on two datasets in Table 6.

### A.3 ALTERNATIVE FORMULATIONS TO LEARN PARAMETERS FOR LOCALITY FEATURES

An alternative way to incorporate locality features into the model is an adaptive variant that conditions the weights and biases ($\{w_n\}, \{b_n\}$ in Equation 6) on the current context representation $f(c_t)$

Table 6: Additional token prediction top-$k$ ($k = 10, 20$) accuracy results and relative error reduction (RER) on two datasets.

| Dataset | Model | Top-10 | RER | Top-20 | RER |
|---|---|---|---|---|---|
| WIKITEXT-103 | Transformer | 72.0% | - | 78.9% | - |
| | +kNN | 74.6% | 9.29% | 81.0% | 9.98% |
| | +kNN + locality feat. | **74.9%** | 1.30% | **81.1%** | 0.84% |
| JAVA GITHUB | Transformer | 89.5% | - | 90.8% | - |
| | +kNN | 97.3% | 74.86% | 98.2% | 80.33% |
| | +kNN + locality feat. | **97.9%** | 21.89% | **98.6%** | 25.41% |

Table 7: The perplexity results comparing alternative formulation using MLP to contextualize parameters for locality features on two datasets.

| Dataset | Model | Dev PPL | Test PPL |
|---|---|---|---|
| WIKITEXT-103 | Transformer | 23.31 | 23.73 |
| | +kNN | 20.21 | 19.94 |
| | +kNN + locality (MLP contextualized) | 20.11 | 19.92 |
| | +kNN + locality (direct) | **19.51** | **19.16** |
| JAVA GITHUB | Transformer | 3.29 | 3.07 |
| | +kNN | 2.43 | 2.18 |
| | +kNN + locality (MLP contextualized) | 2.47 | 2.20 |
| | +kNN + locality (direct) | **2.37** | **2.13** |

parameterized by a MLP:

$$[w_0 \quad ... \quad w_n \quad b_1 \quad ... \quad b_n]^T = MLP(f(c_t)) \tag{8}$$

In our experiments, we used a two-layer MLP with ReLU activations, with 64 hidden units and 0.3 dropout rate during training. The perplexity results compared with directly optimizing weights and biases ($\{w_n\}, \{b_n\}$) are shown in Table 7.

We find that contextualizing the parameters does not result in significant improvements over directly optimizing $w$ and $b$, and sometimes makes the performance even worse. This is perhaps because the context vector space is very large (512-1024 dimensions) compared to the relatively few data points from the validation set used to train.

In Section 6.2, we discuss the effect of learned parameters for each locality level. Observing that the bias terms ($b_i$) and weights ($w_i$) vary according to the locality levels in the learned parameters and to study the weights of the non-local level $w_0$, we freeze all weights except for non-local weights ($w_{i>0}$) to 1 and only optimize bias terms and the weight for the non-local level ($w_0$). This is to exacerbate the effect of bias on different locality levels. The learned parameters are shown in Table 8. We see similar results where the bias terms vary aggressively to modify the "distance" with different levels of locality, and the weights for the non-local level are less than 1, lowering the importance of those non-local retrieved candidates. It's worth mentioning that for JAVA GITHUB these learned biases are much larger in amplitude than before, to compensate for the small scale weights learned before (only around 0.03). However, the perplexity results on both datasets are slightly worse than the full optimization setting that we use in the main experiments (19.33 vs. 19.16 in WIKITEXT-103 and 2.15 vs. 2.13 in JAVA GITHUB).

## A.4 CONNECTION WITH RELATED WORK AND NOVELTY

Previous work (Hellendoorn & Devanbu, 2017) has made the observation that source code files from the same GitHub repository or sub-directory tend to be relatively similar, but did not include an empirical analysis of this effect. Rather, their observation was backed up by an improved performance of their $n$-gram language model with multiple tiered caches. Our work improves on this in a number of ways, including 1. directly examining the internal representations of a neural language model, 2. demonstrating that the internal representations do not sufficiently capture structural locality

Table 8: Learned parameters $\theta_0, \{\theta_n\}$ for each locality level and a non-local level $g_0$, with fixed $w_{i>0} = 1$ during optimization.

|       | WIKITEXT-103 | | JAVA GITHUB | |
|-------|------|--------|------|---------|
|       | $w$  | $b$    | $w$  | $b$     |
| $g_0$ | 1.127 | –     | 0.901 | –      |
| $g_1$ | 1.000 | -0.385 | 1.000 | -28.716 |
| $g_2$ | 1.000 | -0.475 | 1.000 | -55.428 |
| $g_3$ | 1.000 | -0.726 | –    | –       |

features, 3. providing efficient ways to compensate for this disconnect, leading to improved language modeling performance, and 4. showing that this carries over to Wikipedia, which has not been previously examined in this way. As a result, our work both gives more fine-grained insights into this phenomenon and expands the applicability to neural models and new domains. In addition, our work proposes a more generalized formulation for encoding multiple localities across multiple domains than the one proposed in Hellendoorn & Devanbu (2017), which treated locality as strictly nested (e.g. project $\rightarrow$ sub-directory $\rightarrow$ file). Our formulation in Eq. 5 can encode more general hierarchies, such as the *lattice* we used in the Wikipedia case:

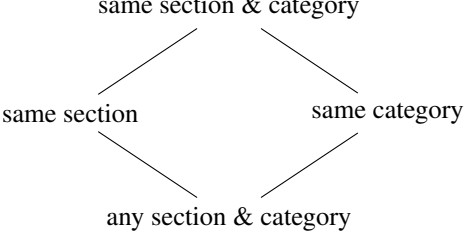

