# OpenReview forum: "Capturing Structural Locality in Non-parametric Language Models"
_ICLR.cc/2022/Conference — ICLR 2022 Poster_

### Official Review · Reviewer_7oBt · 2021-11-02

**Correctness:** 4
**Technical Novelty And Significance:** 2
**Empirical Novelty And Significance:** Not applicable
**Recommendation:** 6
**Confidence:** 3

**Main Review:**

Strengths:
- Authors propose a way to include structural locality into kNN-LM models
- Authors demonstrate improvement on retrieval tasks for Wikipedia and Java source code domains

Weaknesses:
- It seems that the structural locality and its value are known and not new. The paper's contribution then is adding the structural locality to the non-parametric language models. The way to add structural locality is straightforward. It seems to me that this contribution is not significantly novel.
- There is no comparison or discussion of other ways to add structural locality to LMs. It seems authors tried just a single approach that they present in the paper.
- Using locality info in authors' experiments leads to minor improvement. This is not a significant weakness since there might be other tasks where locality will contribute more. However, it would have been interesting to see tasks or domains where improvement is more significant.

Based on discussion with authors and clarifications provided in comments and paper itself, I have increased my score to "marginally above acceptance threshold".

**Summary Of The Paper:**

Authors propose a way to use structural locality in language models. Authors demonstrate that the structural locality information improves results for two domains they experiment with.

**Summary Of The Review:**

In my opinion, the paper makes a minor contribution by proposing a straightforward way to add structural locality to kNN-LM models. The contribution is possibly useful, but not major. For tested tasks, structural locality information improves results but not significantly. I believe the paper is marginally below acceptance threshold.

---

> ### Author Response · Authors · 2021-11-12
> **We thank the reviewer for the detailed comments! We updated the paper and will address your concerns below**
>
> We thank the reviewer for the comments. We have updated the paper to include some ablation studies regarding alternative ways to learn weights for locality features and expanded discussions with empirical results and analysis in Section 6.2 and Appendix A.3. We will address your concerns below.
>
>
> **Novelty**
>
> With respect, we strongly disagree with the statement that the insights provided by this paper are already known in the literature. For example, we believe the analysis of the effect of such locality features in language modeling tasks in Figures 2 and 3 are non-trivial, and we personally learned a lot from this, despite being quite familiar with language modeling in general and non-parametric language models in particular. We also feel that the empirical performance comparisons are of significant value. These methods verify that a) structural locality is not implicitly fully captured by the distance metric used in non-parametric language models and further that b) explicitly plugging in structural locality into non-parametric language models can improve their effectiveness, which c) achieves a new state-of-the-art model for code modeling by a large margin. There was a mention in the review that “structural locality and its value are known and not new.” We were not exactly sure which previous results this was referring to, but to contrast with some other related work: Tu et. al. 2014 only utilizes immediate local context before the predicted tokens, not the structural locality we handle in this work, and it is not clear how these results could be directly applied to neural language models. Hellendoorn & Devanbu 2017 apply similar local context to neural language models for code, but again do not discuss structural locality as we define it. In the text domain, some adaptive LMs (Grave et al., 2016; 2017) use previous hidden states to help predict the next token, but they did not explicitly identify and incorporate locality features. Also, none had analyzed their necessity and utility to non-parametric retrieval-based models. We would be happy to elaborate further if the reviewer could provide any references referring to where these insights have been reported previously.
>
> **Discussion of other ways to add structural locality**
>
> Thank you for the comment, we agree that the results would be stronger with more complete ablations of the different methods with which to incorporate structural locality. We actually had already performed these experiments before submission and discussed in the paper about several alternative ways we attempted to add locality features into the model (Section 5, contextualizing the weights and biases for locality features on current context vector) but ended up not reporting them in detail as they empirically performed worse than the proposed formulation. We revised the paper to include these results in Appendix A.3 for reference.
>
>
> **Minor improvement**
>
> In our opinion, a consistent 3-15% relative error reduction over the very strong KNN-LM baseline across multiple metrics over two datasets from different domains is not so minor, but we realize that subjective opinions may differ here. In addition, it is worth mentioning that our results improve the state-of-the-art on the Java Github code modeling corpus by a large margin; retrieval-based language models had not previously been used in that domain.  But perhaps more importantly (and as also mentioned above), these improvements are founded on both a careful analysis of the workings of this model and involve just a few added parameters that are also highly interpretable, and thus the encouraging results here may spur methods that tackle a similar problem in a different way, resulting in further improvements as well.

---

> > ### Comment · Reviewer_7oBt · 2021-11-18
> > **Response to author comments**
> >
> > Thank you for your extensive comments.
> > Regarding the novelty of the structural locality: I thought that the paper said that the structural locality properties were known. From your response, it seems that you are saying that structural locality was not known or not used before. Is this the case? If you claim novelty of the structural locality concept and/or its expression for Wikipedia and Github domains, then you should make this explicit. If not, then I think my original comment stands.
> > Regarding how the structural locality is added to the model, it seems to me that it is straightforward to think about structural locality as an extra distance metric. I agree that learning that additional function is somewhat novel and that comparing to alternative ways of adding it is useful.
> >
> > Considering all information, I am OK if this paper is accepted to the conference.
> >
> > Thank you.

---

> > > ### Author Response · Authors · 2021-11-18
> > > **Thank you for the quick reply! We further clarify your concerns below**
> > >
> > > Thank you for the follow-up; we understand the confusion now. Previous work (Hellendoorn and Devanbu, 2017) did make the observation that source code files from the same GitHub repository or subdirectory tend to be relatively similar, but did not include an empirical analysis of this effect; rather, they showed that this is useful for the performance of cache-based (n-gram) language models. We (1) directly examine the internal representations of a neural language model, (2) demonstrate that the internal representations do not sufficiently capture structural locality features, (3) demonstrate ways to compensate for this disconnect, leading to improved language modeling performance, and (4) also demonstrate that this carries over to Wikipedia, which has not been previously examined in this way. This both gives more fine-grained insights and expands the applicability to neural models and new domains. In addition, our work proposes a more generalized formulation for encoding *multiple* localities across multiple domains. Previous work treats locality as strictly nested (e.g. project->subdirectory) while ours can encode more general hierarchies such as the lattice we used in the Wikipedia case (e.g. same category - yes/no combined with same section - yes/no).
> > >
> > > If this explanation seems clear we can revise the paper to more clearly reflect this discussion of the difference between our work and the previous work. Thank you again.

---

> > > > ### Comment · Reviewer_7oBt · 2021-11-19
> > > > **Re clarification**
> > > >
> > > > Yes, this would clarify the contribution and differences from the past work on structural locality.
> > > >
> > > > A comment for future work: as couple reviewers have observed, structural locality may have to be defined differently for different domains. It might be interesting to look wider in which domains and what structural locality metrics are useful (i.e. not captured in simple distance metrics) and which ones are already captured in distance. Also what generalizable definitions of structural locality and its metrics can be created. Looking even wider, structural locality should be applicable to non language domains, for example, image retrieval where structural locality could be similar to proposed here (images from single user or single album to be closer vs others) or possibly different. Not something for this paper, but in case there is interest to pursue this further.
> > > >
> > > > Thank you.

---

> > > > > ### Author Response · Authors · 2021-11-21
> > > > > **Follow-up and Revised the paper to incorporate the discussion**
> > > > >
> > > > > Thank you for your feedback. We have incorporated this discussion in the paper in a new footnote (footnote 9) in the “Conclusion” section and direct the reader to more details presented in the newly added Appendix A.4, which explains the relation to previous work and our core novelties in more detail. We hope that these changes improve the quality of the paper. Please let us know of any further suggestions for clarifications that would improve your view of our work.
> > > > >
> > > > > Regarding your comment on future work: we appreciate your ideas of these future studies. The idea of locality metrics that go beyond modifying distance is quite interesting. This echoes our own position of the paper's implications as well; for instance as mentioned in our "Limitations" paragraph at the very end of the paper, we think it would be interesting to see how this work performs on even more domains.

---

> > > ### Author Response · Authors · 2021-11-21
> > > **Updated paper incorporating the discussion**
> > >
> > > Thank you for your feedback. We have incorporated this discussion in the paper in a new footnote (footnote 9) in the “Conclusion” section and direct the reader to more details presented in the newly added Appendix A.4, which explains the relation to previous work and our core novelties in more detail. We hope that these changes improve the quality of the paper. Please let us know of any further suggestions for clarifications that would improve your view of our work.
> > >
> > > Regarding your comment on future work: we appreciate your ideas of these future studies. The idea of locality metrics that go beyond modifying distance is quite interesting. This echoes our own position of the paper's implications as well; for instance as mentioned in our "Limitations" paragraph at the very end of the paper, we think it would be interesting to see how this work performs on even more domains.

---

### Official Review · Reviewer_gYwe · 2021-11-02

**Correctness:** 4
**Technical Novelty And Significance:** 3
**Empirical Novelty And Significance:** 3
**Recommendation:** 8
**Confidence:** 3

**Main Review:**

+ The paper is well written and easy to read
+ What the paper is suggesting is simple, yet useful. The example depicted in Figure 1 describes their motivation as well as what they are doing effectively
+The concept of structural locality as used in their paper is defined clearly
+Results are compared with other state of the art models (Table 2)

- The only weakness that I can identify is that the authors have used one dataset of each domain. Since there is minor improvement in the results, it is difficult to gain confidence that the results will indeed improve on other datasets as well






**Summary Of The Paper:**

The authors propose an approach to complement context by adding 'locality' information in examples present in external stores of non-parametric language models. The locality information captures the hierarchical structure, and deems two contexts more similar (or having less distance) if they share common hierarchical structure/attributes. The authors conduct experiments on source code as well as natural language articles. They analyze the results to point out reasons for improvement, and also highlight differences between the domains.

**Summary Of The Review:**

Same as above

---

> ### Author Response · Authors · 2021-11-12
> **We thank the reviewer for the detailed comments! We updated the paper and will address your concerns below**
>
> We thank the reviewer for the comments and for the appreciation of our work! We have updated the paper to include some ablation studies regarding alternative ways to learn weights for locality features and expanded discussions with empirical results and analysis in Section 6.2 and Appendix A.3. We will address your concerns below.
>
> **Only one dataset of each domain**
>
> We understand the potential concern, but at the same time think that consistent results across both English and Java Source Code demonstrate a relatively high level of generality already. We would be happy to have any concrete suggestions for other datasets that would be most appropriate to test on though.

---

### Official Review · Reviewer_bn3A · 2021-11-02

**Correctness:** 4
**Technical Novelty And Significance:** 2
**Empirical Novelty And Significance:** 1
**Recommendation:** 6
**Confidence:** 2

**Main Review:**

The paper models structural information into the non-parametric language models. While the results demonstrate that the method improves upon the existing methods, there are some weaknesses too.

Strengths:
1. The model or the loss function developed by the authors that incorporate structural information is novel. The authors have also clearly explained the model.
2. Results demonstrate that the method improves upon existing methods.

Weaknesses:
1. While there are clear strengths, one weakness is that one may need to define structural properties in different types of datasets that one might use. For instance, it is clear that the model works for source codes and Wikipedia because associated structural information can be mined from the data. It is unclear how does the method generalise across different tasks and datasets, i.e., beyond two datasets. While the authors have addressed these limitations towards the end of the paper, the question is will the work be useful only to a small set of audience, or people from different domains can manually or automatically build such prior knowledge and incorporate it in this model. The key advantages are clear from the paper, this seems to be the weakness that is hard to defend. One possible way to improve the argument so that we could obtain Wikidata-type structure for most datasets is to exploit entity detection and linking including automatically learning their relation (vector) information in a completely unsupervised way. The authors must note that I am simply giving ideas on how to strongly defend this weakness of the model.
2. In terms of experiments, these can be further improved by conducting some downstream application tasks. Can the model be useful for document classification tasks? Currently, it is very difficult to gauge the usefulness of the model through the limited experiments presented in the paper which mainly revolves around perplexity analysis and table 5 in the appendix has additional token prediction results.


**Summary Of The Paper:**

The paper is about modelling structural locality in non-parametric language models. The key hypothesis is in modelling not only the co-occurrence characteristics but also structural characteristics such as locality. The paper explains the key claims via case studies conducted on source code data and Wikipedia datasets.

The model paradigm is based on non-parametric language models. A key difference between the non-parametric model and the parametric counterpart is that in the non-parametric model the model parameters are not only determined by the model architecture but also the underlying data.

Structural locality, which is different from just co-occurrence counts, models the structural relationships between pairs of items, e.g., whether they belong to the same or different directory in the case of source code.

The optimisation model is presented in Equation 7 where the authors need a small sample set emanating from the same domain to train the model. The authors then conduct experiments to demonstrate that the method improves upon existing works. Both qualitative and quantitative experimental results are shown.

**Summary Of The Review:**

Overall, the paper indeed has some merits. The paper can be made stronger by considering some comments mentioned above.

---

> ### Author Response · Authors · 2021-11-12
> **We thank the reviewer for the detailed comments! We updated the paper and will address your concerns below**
>
> We thank the reviewer for the detailed comments! We have updated the paper to include some ablation studies regarding alternative ways to learn weights for locality features and expanded discussions with empirical results and analysis in Section 6.2 and Appendix A.3. We will address your concerns below.
>
> **The need to define structural properties**
>
> We would like to first state that both Wikipedia and GitHub are huge and popular datasets already, and the recent GitHub CoPilot/OpenAI Codex is also trained with a plethora of data from Github repositories. In addition, there is a lot of data in the real world that directly follows the same format as Wikipedia (all other Wiki-type/structured documents) and Github (all source code repositories).
>
> Further, we agree that it would be ideal if a language model was able to learn how to capture structural locality without explicit feature design. However, one major empirical result of our study is that embeddings of current models do *not* perfectly capture structural locality, as Figures 2 and 3 demonstrate. We argue that this is not a weakness of our method, but rather a weakness of the baseline method, and pointing out this weakness is one of the contributions of our paper. The specific method we employed to mitigate this problem in the current study requires custom locality features, but on a higher level we believe our work demonstrates the importance of considering the structural properties of datasets in non-parametric language models, and we hope that it will also spur future work on alternative methods to tackle this problem as well.
>
> Last but not the least, we greatly appreciate your suggested idea of using Wikidata information to do entity detection for other contexts. We think it’s clever and we would like to consider this as future work, but it might be tough to fit in the current paper given the space limitations. One potential concern is that these methods may be prone to error propagation given that the entity linking may not be perfect in many contexts, but mitigation strategies for this could certainly be devised as well. In this paper, we mainly focus on “user-defined” prior knowledge that could be obtained directly through user input or readily available metadata for datasets, which already covers a large variety of domains.
>
> **Downstream application tasks**
>
> Thank you for the comment! We actually did include one downstream task metric, predictive accuracy (at 1 and at 5), which is the standard evaluation metric for code completion [1, 2, 3] and predictive text entry [4]. Upon a second reading of the paper we realize this was not very clear, so we have revised the paper to reflect this. Other downstream tasks such as document classification would also be excellent to add, but as far as we know using these tasks to evaluate non-parametric language models such as KNN-LM is not standard in the literature, and we don’t have a very clear idea how we could do so. Any suggestions on an appropriate experimental setup would be welcome!
>
> [1] Raychev, V., Vechev, M., & Yahav, E. (2014, June). Code completion with statistical language models. In Proceedings of the 35th ACM SIGPLAN Conference on Programming Language Design and Implementation (pp. 419-428).
>
> [2] Alon, U., Sadaka, R., Levy, O., & Yahav, E. (2020, November). Structural language models of code. In International Conference on Machine Learning (pp. 245-256). PMLR.
>
> [3] Karampatsis, Rafael-Michael, et al. "Big code!= big vocabulary: Open-vocabulary models for source code." 2020 IEEE/ACM 42nd International Conference on Software Engineering (ICSE). IEEE, 2020.
>
> [4] Stocky, T., Faaborg, A., & Lieberman, H. (2004, April). A commonsense approach to predictive text entry. In CHI'04 Extended Abstracts on Human Factors in Computing Systems (pp. 1163-1166).

---

> > ### Comment · Reviewer_bn3A · 2021-11-16
> > **Response to author comments.**
> >
> > I thank the authors for responding to my comments. I also thank the authors for including a few more details on the experiments' side. I am convinced that the datasets used by the authors are sufficient enough to conduct a reliable study. My original comment mainly revolved around the fact that there are other interesting use cases, but I do agree that not everything could be done in a limited-page paper.
> >
> > Since the authors have addressed my comments convincingly, I am tempted to change my recommendation score.

---

> > > ### Author Response · Authors · 2021-11-16
> > > **Thank you for your prompt reply!**
> > >
> > > I thank you for the prompt reply and considerations regarding our revision and responses. I am glad that the conversation clears things up.

---

### Official Review · Reviewer_PnXs · 2021-11-03

**Correctness:** 4
**Technical Novelty And Significance:** 2
**Empirical Novelty And Significance:** 2
**Recommendation:** 6
**Confidence:** 4

**Main Review:**

This work concerns itself about utilizing structural locality inherent in real-world datasets in improving the effectiveness of non-parametric language models. It makes a claim that a) structural locality is not implicitly fully captured by the distance metric used in non-parametric language models and further that b) explicitly plugging in structural locality into non-parametric language models can improve their effectiveness. It validates this claim first by doing analysis of two datasets with the help of custom locality functions and then by plugging in the locality functions into a non-parametric language model with the help of learnable parameters.


Positives:
	1. Well stated hypothesis and analysis that shows that structural locality is not implicitly fully captured by the distance metric used in K-nearest neighbour non-parametric language model of Khandelwal et al.
	2. Locality features for two datasets - wikipedia and Java projects
	3. Incorporation of locality features in non-parametric language models using simple learnable functions of the distance metric
	4. Analysis that shows that incorporating locality features leads to improved distance distribution among nearest neighbours

Negatives:
	1. Structural locality inherent in datasets need to be captured by a set of custom locality feature functions which requires prior knowledge of the domain of the datasets.
	2. Marginal improvements in results
	3. No detailed discussion of learned parameters presented in Table 3.
		a. It appears that w doesn't matter so much for all non l_0 features. So it would be interesting to set w to 1 for these features and learn only b and for l_0 learn w.

In Eqn 4. y should be w_t?



**Summary Of The Paper:**

This work concerns itself about utilizing structural locality inherent in real-world datasets in improving the effectiveness of non-parametric language models. It makes a claim that a) structural locality is not implicitly fully captured by the distance metric used in non-parametric language models and further that b) explicitly plugging in structural locality into non-parametric language models can improve their effectiveness. It validates this claim first by doing analysis of two datasets with the help of custom locality functions and then by plugging in the locality functions into a non-parametric language model with the help of learnable parameters.


**Summary Of The Review:**

The paper makes an interesting hypothesis and goes about validating the hypothesis. However, the improvements due to the proposed method are marginal.

---

> ### Author Response · Authors · 2021-11-12
> **We thank the reviewer for the detailed comments! We updated the paper and will address your concerns below**
>
> We thank the reviewer for the detailed comments! We have updated the paper to include some ablation studies regarding alternative ways to learn weights for locality features and expanded discussions with empirical results and analysis in Section 6.2 and Appendix A.3. We will address your concerns below.
>
> **Structural locality needs to be captured by a set of custom locality feature functions which requires prior knowledge of the domain of the datasets**
>
> We agree that it would be ideal if a language model was able to learn how to capture structural locality without explicit feature design. However, one major empirical result of our study is that embeddings of current models do *not* perfectly capture structural locality, as Figures 2 and 3 demonstrate. We argue that this is not a weakness of our method, but rather a weakness of the baseline method, and pointing out this weakness is one of the empirical contributions of our paper. The specific method we employ to rectify this weakness in the current study requires custom locality features, but on a higher level we believe our work demonstrates the importance of considering the structural properties of datasets in non-parametric language models, and we hope that it will also spur future work on alternative methods to tackle this problem as well.
>
> **Marginal improvements in results**
>
> In our opinion, a consistent 3-15% relative error reduction over the very strong KNN-LM baseline across multiple metrics over two datasets from different domains is not “marginal”, but we realize that subjective opinions may differ here. In addition, it is worth mentioning that our results improve the state-of-the-art on the Java Github code modeling corpus by a large margin; retrieval-based language models have not been previously used in that domain.  But perhaps more importantly (and as also mentioned above), these improvements are founded on both a careful analysis of the workings of this model and involve just a few added parameters that are also highly interpretable. Thus, the encouraging results here may spur methods that tackle a similar problem in a different way, resulting in further improvements as well.
>
> **Discussion of learned parameters presented in Table 3**
>
> Thank you for the comment! We agree that a more complete discussion of this is useful.
> In the original submitted paper, there was already a discussion in Section 6.2 regarding the learned parameters, “the biases vary strongly, as the model tries to “correct” the original distance by emphasizing more local contexts”. The effect of the learned parameters was also illustrated by comparing the “distance” before and after taking into account the weighted locality features, in Figure 3, which we described as “Compared with the original negative distance…, the negative modified distance is more separated between the different locality levels on either dataset, showing the relative importance of different locality more clearly.”
> To dig a bit deeper than what was included in the originally submitted version, we expanded the discussion in Section 6.2 and Appendix A.3 of the revised paper, specifically following the suggestion to fix all “w” except for w0 to 1 and optimize only w0 and bias terms b_n. We found that for the Wiki dataset, the learned parameters were w0 = 1.1267 and b1 = -0.3851, b2 = -0.4745, b3=-0.7261, which is in line with the discussion made in the paper where the bias varies to modify the “distance” with different levels of locality. The final perplexity on the test set was slightly worse than the full setting in the paper (19.33 vs 19.16).
> For the Java dataset, we observed a similar effect: the learned parameters were w0 = 0.9017 and b1 = -28.7163, b2 = -55.4277. It’s worth mentioning that these biases are way larger in amplitude than before, to compensate for the small weights learned originally (~0.03). The perplexity is also slightly worse: 2.15 vs. 2.13.
>
> **Typo**
>
> Thanks for pointing out the typo in Eq. 4 and we have corrected this in the revised paper.

---

> > ### Comment · Reviewer_PnXs · 2021-11-17
> > **Discussion of learned parameters**
> >
> > Thank you for implementing my suggestion to fix all “w” except for w0 to 1 and optimize only w0 and bias terms b_n". This highlights the relative importance of different locality functions in a much better way.

---

> > > ### Author Response · Authors · 2021-11-17
> > > **Thank you for the quick reply**
> > >
> > > Thank you for the quick reply and for your suggestion. In case you missed it, we have also uploaded the revised paper to incorporate your suggestions in the appendix while mentioning the effects and alternatives in the main text due to page limits. Happy to revise again in case things are not clear in the revised version of the paper.

---

> > > > ### Author Response · Authors · 2021-11-21
> > > > **New revision updates**
> > > >
> > > > Following the discussion with Reviewer 7oBt, we have incorporated a more detailed discussion, which explains the relation to previous work and our core novelties in more detail, in footnote 9 in the “Conclusion” section and direct the reader to more details presented in the newly added Appendix A.4. We hope that these changes improve the quality of the paper.
> > > > Please let us know of any further suggestions for clarifications that would improve your view of our work.

---

> > > ### Author Response · Authors · 2021-11-21
> > > **Additional Revision Updates**
> > >
> > > Following the discussion with Reviewer 7oBt, we have incorporated a more detailed discussion, which explains the relation to previous work and our core novelties in more detail, in footnote 9 in the “Conclusion” section and direct the reader to more details presented in the newly added Appendix A.4. We hope that these changes improve the quality of the paper. Please let us know of any further suggestions for clarifications that would improve your view of our work.

---

### Author Response · Authors · 2021-11-12
**Paper revisions**

We thank the reviewers for their insightful feedback and appreciating our work! We have updated the paper to include some ablation studies regarding alternative ways to learn weights for locality features and expanded discussions with empirical results and analysis in Section 6.2 and Appendix A.3.

In more detail, we first addressed Reviewer PnXs’s comment on an alternative way to formulate the parameter learning for locality features, and included results in Appendix A.3. We found that this led to similar observations with respect to the learned parameters but found that it caused slightly worse model performance.

Secondly, we addressed Reviewer 7oBt’s suggestion regarding having different methods with which to incorporate structural locality by adding complete results of alternatives we tried but ended up not using because they empirically performed worse than the proposed formulation, also in Appendix A.3.

---

> ### Author Response · Authors · 2021-11-21
> **Revision updates**
>
> Following the discussion with Reviewer 7oBt, we have incorporated a more detailed discussion, which explains the relation to previous work and our core novelties in more detail, in footnote 9 in the “Conclusion” section and direct the reader to more details presented in the newly added Appendix A.4. We hope that these changes improve the quality of the paper.

---

### Comment · Reviewer_PnXs · 2021-11-25
**Response after reading all reviews, responses and revised submission**

I have gone through the reviews by other reviewers, the responses by the authors and the revised submission.

I thank the authors for implementing the following suggestion made by me in my original review:
"It appears that w doesn't matter so much for all non l_0 features. So it would be interesting to set w to 1 for these features and learn only b and for l_0 learn w."
The experimental study done by the authors in the context of this suggestion seem to confirm my hypothesis. One positive implication of this is that fewer parameters need to be learned while not sacrificing the gains (section A.3). Further, the weights seem to be more interpretable and can provide a heuristic for determining the relative importance of different locality features.

As noted in my original review, there are several positives in this submission and I continue to be appreciative of them. The submission is well written and the engagement of the authors to the reviews has been positive. The main drawback of this work is that it doesn't provide a principled procedure or heuristics for formulating effective locality features for different datasets. The task of formulating the locality features remain outside the scope of the current work and needs domain knowledge. This limits the potential impact of this work as also noted by other reviewers. All that can be concluded from the experimental study is that the carefully designed locality features seem to give minor improvement over language models that don't use such features.

I'm willing to upgrade my recommendation to marginally above acceptance threshold.

---

> ### Comment · Reviewer_7oBt · 2021-11-25
> **Changed score to "marginally above acceptance threshold"**
>
> Based on discussion with authors and clarifications in comments and paper itself, I have changed my score to "marginally above acceptance threshold" and noted that in my review.

---

> > ### Author Response · Authors · 2021-11-26
> > **Thank you for the appreciation and suggestion**
> >
> > I thank you for the prompt reply and considerations regarding our revision and responses. I am glad that the conversation clears things up and the revisions improved the paper. I am also appreciative of your other comments and will seriously consider them as future work.

---

> ### Author Response · Authors · 2021-11-26
> **Thank you for the appreciation and suggestion**
>
> I thank you for the prompt reply and considerations regarding our revision and responses. I am glad that the conversation clears things up and the revisions improved the paper. I am also appreciative of your other additional suggestions outside the scope of this paper and will seriously consider them as future work.

---

### Decision · Program_Chairs · 2022-01-20

**Decision:**

Accept (Poster)

**Comment:**

Reviewers were in agreement but borderline.  The paper has a nice hypothesis and develops the work using two realistic datasets, Wikipedia and Code.  One reviewer was initially more negative but changed their views based on the authors improvements to the paper.
The idea is fairly simple, but does require modellers come up with the structural features.  There was discussion that more down-stream tasks are needed to highlight the approach.  Moreover, more datasets should be experimented with.  In all, experiments are good but improvement is easily done.